# STRATEGIC GENERALIZATION WITHOUT INTERACTION: CAN POST-TRAINING ALONE INDUCE MULTI-AGENT BEHAVIOR?

## ABSTRACT

Directly training Large Language Models (LLMs) for Multi-Agent Systems (MAS) remains challenging due to intricate reward modeling, dynamic agent interactions, and demanding generalization requirements. This paper explores whether post-training techniques can effectively generalize to multi-agent scenarios *without any interactive multi-agent data*. We use economic reasoning as a testbed, leveraging its strong foundations in mathematics and game theory, its demand for structured analytical reasoning, and its relevance to real-world applications such as market design, resource allocation, and policy analysis. We introduce **Recon** (Reasoning like an ECONomist), a 7B-parameter open-source LLM post-trained on a hand-curated dataset of 2,100 high-quality economic reasoning problems. Comprehensive evaluations show that Recon substantially improves economic reasoning benchmarks and generalizes to unseen multi-agent games, exhibiting equilibrium-seeking behavior. To our knowledge, this is the first systematic study to demonstrate that domain-aligned post-training can induce emergent strategic behavior in multi-agent settings. These findings underscore post-training as a scalable route to structured reasoning and agent alignment, shedding light on the roles of SFT and RL in cultivating emergent behaviors.

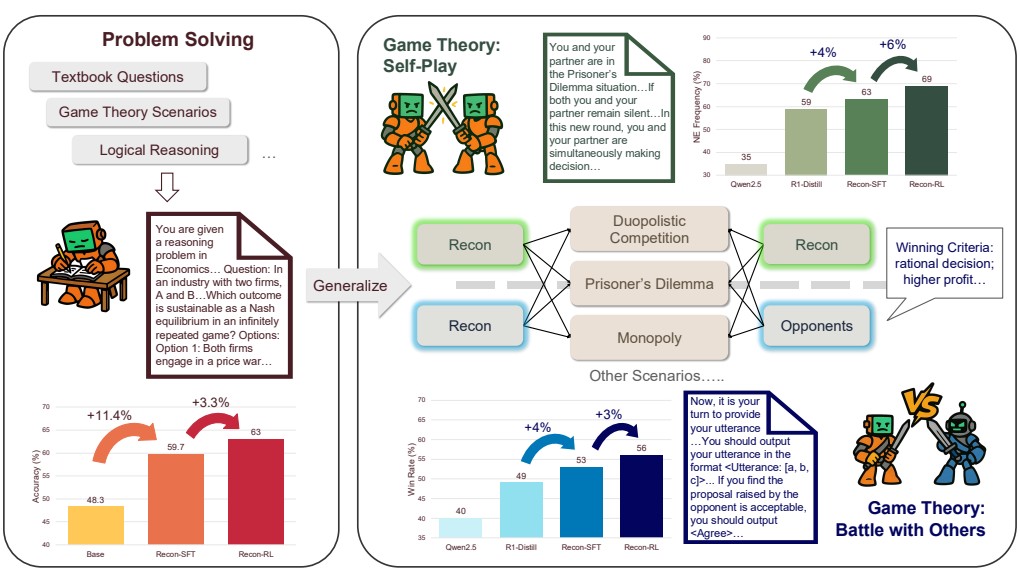

Figure 1: **Overview of Recon.** Post-training on curated economic reasoning tasks enables large language models to generalize from textbook-style problems to interactive game-theoretic settings. Recon improves accuracy on economic benchmarks and exhibits emergent strategic behavior, achieving higher Nash equilibrium convergence and win rates in unseen multi-agent games.

# 1 INTRODUCTION

Large Language Models (LLMs) have recently progressed from general-purpose text generation to exhibiting strong reasoning capabilities across mathematics and coding, as exemplified by OpenAI's o1 series (OpenAI, 2024b) and DeepSeek-R1 (DeepSeek-AI, 2025). This transition has been driven by techniques such as Chain-of-Thought (CoT) prompting, Supervised Fine-Tuning (SFT), and Reinforcement Learning from Human Feedback (RLHF) (Wei et al., 2022; Kojima et al., 2022; Ouyang et al., 2022; Lightman et al., 2023), culminating in the emergence of Large Reasoning Models (LRMs) (Chen et al., 2025a; Xu et al., 2025a). A key framework in this space is Reinforcement Learning with Verifiable Rewards (RLVR) (Lambert et al., 2025), which replaces standard reward models with outcome-verification functions for tasks such as math solving and instruction following. RLVR has since been extended to Medicine (Zhang et al., 2025a), SQL (Ma et al., 2025), Logic (Xie et al., 2025), and Finance (Liu et al., 2025b; Qian et al., 2025; Zhu et al., 2025). Complementary methods such as Su et al. (2025) and Liu et al. (2025d) extend RLVR to soft or online reward signals, while LIMO (Ye et al., 2025), LIMR (Li et al., 2025), and s1 (Muennighoff et al., 2025) demonstrate that post-training can elicit strong reasoning in smaller models under specialized, limited data.

In parallel, LLM-based Multi-Agent Systems (MAS) have gained prominence as platforms for exploring complex interactions, cooperation, and emergent social behaviors (Park et al., 2023; Zhou et al., 2024; Li et al., 2023). A pivotal objective within MAS is economic rationality—the capability to systematically reason about incentives, trade-offs, and strategic decision-making—which underpins effective coordination and negotiation. The STEER benchmark (Raman et al., 2024; 2025) formalizes economic rationality by testing LLMs on foundational principles such as utility maximization, behavioral bias, and strategic reasoning. This aligns closely with game theory, a longstanding theoretical foundation for MAS research (Cesa-Bianchi & Lugosi, 2006; Zhang et al., 2021; Slumbers et al., 2023; Mazumdar et al., 2025), increasingly central to evaluating LLM-based agents (Xu et al., 2024; Zhang et al., 2024; Fan et al., 2024; Sun et al., 2025). Several recent studies have focused on the reasoning abilities of LLM agents in these settings, highlighting both their importance and their limitations (Piedrahita et al., 2025; Jia et al., 2025; Zhang et al., 2025b). Ongoing efforts to develop unified economic-agent environments and benchmarks (Li et al., 2024; Duan et al., 2024; tse Huang et al., 2025; Hua et al., 2024) further reinforce the centrality of this research direction.

Despite significant interest, directly training LLMs for multi-agent interactions remains complex and underexplored, often hampered by challenges like dense reward modeling, unstable coordination dynamics, and conflicting agent objectives (Du et al., 2025). Existing methods, such as multi-agent co-training (Yue et al., 2025a) and MARFT (Liao et al., 2025), typically require extensive supervision and tailored agent architectures, limiting their scalability and generalization potential. This prompts a critical research question:

*Can post-training techniques generalize effectively to multi-agent scenarios?*

To our knowledge, this work is the first to pose and systematically investigate whether post-training alone—without any interactive gameplay data—can induce multi-agent behavior (Figure 1). We adopt economic reasoning as a testbed, given its structured mathematical foundations and strategic dynamics essential to multi-agent systems. Economic tasks frequently involve intricate multi-step reasoning, such as evaluating trade-offs, aligning incentives, and anticipating others' behaviors—ideal for leveraging improvements from SFT and RLVR. While previous studies primarily *assess* economic rationality (Raman et al., 2024; 2025; Hua et al., 2024; Duan et al., 2024; tse Huang et al., 2025), our work actively *enhances* it via targeted post-training. Additionally, real-world applications reinforce this domain's significance, demonstrated by simulations of heterogenous economic-agent roles using LLMs (Hao & Xie, 2025; Li et al., 2024; Xiao et al., 2025).

In this paper, we introduce **Recon**, an LLM specifically designed for structured economic decision-making. We curate a high-quality dataset comprising 2,100 examples spanning 15 critical economic categories, including behavioral bias detections, repeated-game strategies, mechanism-design equilibria. This dataset builds upon and expands benchmarks such as STEER (Raman et al., 2024), EconLogicQA (Quan & Liu, 2024), and EconNLI (Guo & Yang, 2024). Recon employs Supervised Fine-Tuning (SFT) and subsequent Group Relative Policy Optimization (GRPO) (Shao et al., 2024), fine-tuning the DeepSeek-R1-Distill-Qwen-7B model (DeepSeek-AI, 2025; Yang et al., 2024) to enhance structured reasoning and test generalization capabilities across both single-agent and multi-agent economic tasks.

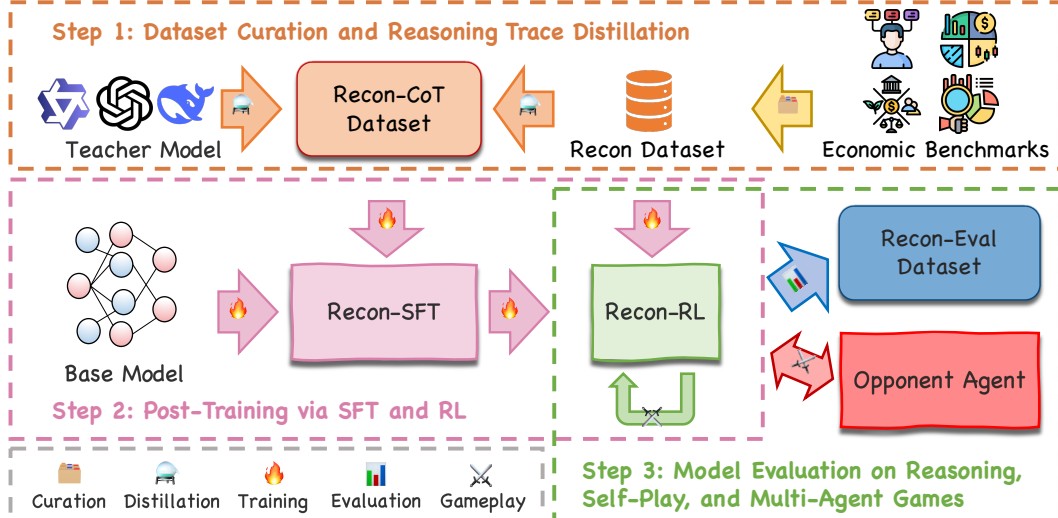

Figure 2: **Pipeline of Recon. Step 1**: We curate a high-quality economic dataset (Recon Dataset) from benchmarks such as STEER, and distill reasoning traces from teacher models to construct the Recon-CoT Dataset. **Step 2**: A base model is post-trained via supervised fine-tuning (Recon-SFT) on Recon-CoT and reinforcement learning (Recon-RL) on the Recon Dataset. **Step 3**: The resulting models are evaluated on reasoning benchmarks (Recon-Eval Dataset), self-play, and multi-agent games against opponent agents.

Our experimental results demonstrate clear improvements in structured reasoning and strategic decision-making through domain-aligned post-training. Notably, models trained on economic problems display economically rational behavior in multi-agent games, despite receiving no interaction-based supervision. This suggests that structured problem-solving can promote latent alignment with game-theoretic principles, indicating that post-training not only enhances task-level accuracy but also encourages emergent rational behavior. These findings provide fresh insights into the distinct roles of SFT and RL in shaping model behavior, generalization, and alignment (Yue et al., 2025b; Chu et al., 2025; Wang et al., 2025e; Liu et al., 2025c; Gandhi et al., 2025). Our contributions include:

- We curate a high-quality dataset of 2,100 problems across 15 economic reasoning categories designed to assess core rationality skills.
- We introduce Recon, a 7B open-source model post-trained via SFT and GRPO for structured economic and strategic reasoning.
- We empirically show that reasoning-oriented post-training enhances both benchmark accuracy and generalization to unseen multi-agent settings.
- We hypothesize post-training as a scalable route to agent alignment, where economic problem-solving effectively fosters strategic behavior.

## 2 RELATED WORK

**Economic Agent Applications.** Recent work has integrated LLMs into economic simulations and agent-based modeling across a range of applications. Hao & Xie (2025) proposed a multi-agent LLM framework for policy analysis, simulating heterogeneous societal groups. Li et al. (2024) introduced EconAgent for macroeconomic modeling, demonstrating human-like decision-making in LLM-driven agents. Xiao et al. (2025) developed TradingAgents to model financial markets with specialized roles such as analysts and traders. Wu et al. (2025) applied LLMs to generate persuasive, context-grounded marketing content for real estate. Lazebnik & Shami (2025) combined LLMs with reinforcement learning to simulate tax evasion dynamics, and Yu et al. (2024) introduced FINCON, a synthesized multi-agent system using conceptual verbal reinforcement for financial decision-making. While these works focus on application design, our approach complements them by enhancing

economic reasoning and decision-making capabilities through post-training—potentially improving performance in real-world multi-agent settings.

**Game-Theoretic Evaluation.** Game-theoretic reasoning has become an essential evaluation paradigm for assessing LLM performance in multi-agent scenarios. Benchmarks such as GT-Bench (Duan et al., 2024), GameBench (Costarelli et al., 2024), and GAMABench (tse Huang et al., 2025) evaluate strategic reasoning across cooperative, adversarial, and sequential games. Several studies focus on negotiation: LAMEN (Davidson et al., 2024) and Abdelnabi et al. (2024) examine stakeholder deliberation, while Hua et al. (2024) introduce a formal agent workflow for modeling negotiation games and equilibrium behavior. GLEE (Shapira et al., 2024) provides a unified benchmark for economic interactions, and Akata et al. (2025) study repeated games to analyze long-term cooperation. Piedrahita et al. (2025) highlight the challenge that increased reasoning capacity in LLMs can, seemingly paradoxically, undermine cooperation, especially in public goods settings. Jia et al. (2025) further show that strategic performance depends more on reasoning quality than scale, and that CoT prompting is not a universal enhancer. Zhang et al. (2025b) emphasize that both metacognitive and strategic reasoning are essential for agent success in real-world, incomplete-information settings like labor markets. Unlike these evaluation-oriented studies, our research leverages post-training techniques to actively enhance LLMs' reasoning abilities and generalize their strategic decision-making to broader economic and multi-agent contexts.

## 3 METHODOLOGY

### 3.1 OVERALL PIPELINE

Our training pipeline comprises two core post-training stages: supervised fine-tuning (SFT) on synthetic reasoning data, followed by reinforcement learning with verifiable rewards (RLVR) on curated economic problems. We detail the dataset curation process in Section 4. Figure 2 provides a schematic overview of the full pipeline, illustrating the flow from data generation to post-training and downstream multi-agent gameplay.

### 3.2 BASE MODEL

We select **DeepSeek-R1-Distill-Qwen-7B**[1] (Yang et al., 2024) as our base model due to its strong reasoning ability, inherited from DeepSeek-R1 (DeepSeek-AI, 2025) through targeted distillation. Among open-source models, Qwen-based (Yang et al., 2024) variants consistently outperform comparable LLaMA-based (Grattafiori et al., 2024) counterparts on multiple benchmarks. The 7B parameter scale offers a practical balance between performance and efficiency, making it well-suited for fine-tuning. Furthermore, DeepSeek-R1-Distill-Qwen-7B achieves competitive results, surpassing GPT-4o (OpenAI, 2024a) on challenging reasoning benchmarks such as MATH (Hendrycks et al., 2021) and AIME (MAA, 2024), thus providing a robust foundation for further adaptation.

### 3.3 POST-TRAINING ALGORITHMS

We use SFT and GRPO as our post-training techniques. SFT aligns the model to structured reasoning traces distilled from teacher models, providing stable initialization. GRPO further optimizes the model via Reinforcement Learning with Verifiable Rewards, encouraging generalization in multi-agent settings. We leave detailed descriptions to Appendices A.2 and A.3.

### 3.4 REWARD DESIGN

To support structured outputs during GRPO training, we develop a hierarchical, rule-based reward function. It scores responses across three stages, format validity, answer extraction, and correctness, while enforcing consistent use of `<think>` and `\boxed{}` conventions to align reasoning traces with final predictions. Full logic and implementation details are provided in Appendix A.4.

---

[1] https://huggingface.co/deepseek-ai/DeepSeek-R1-Distill-Qwen-7B

## 4 DATASET CURATION

Table 1: Accuracy of each model across economic reasoning categories (detailed in Appendix A.5).

| Model Names | Mathematical Foundations | Single-Agent Enviornments | Multi-Agent Enviornments | Representing Other Agents | Logical Reasoning |
|---|---|---|---|---|---|
| R1-Distill-Qwen-7b | **0.896** | 0.751 | **0.630** | 0.740 | 0.430 |
| Qwen-2.5-7b-Instruct | 0.875 | 0.775 | 0.514 | 0.680 | **0.650** |
| Llama-3.1-8b-Instruct | 0.619 | 0.637 | 0.309 | 0.569 | 0.350 |
| Gemma-2-9b-it | 0.828 | 0.798 | 0.375 | 0.639 | 0.590 |
| GPT-4o | 0.887 | **0.841** | 0.625 | **0.782** | 0.640 |
| GPT-3.5-turbo | 0.593 | 0.756 | 0.377 | 0.608 | 0.510 |

Table 2: Accuracy on specified question types from *Multi-Agent Environments* category (detailed in Appendix A.5).

| Model Names | Back. Ind. | Bayes Nash | Best Resp. | Dom. Strat. | Dom'd Strat. | Enf. | Feas. | Intp. | Iter. Rem. | Pure Nash | Subg. Nash | Trig. |
|---|---|---|---|---|---|---|---|---|---|---|---|---|
| R1-Distill-Qwen-7b | 0.360 | **0.681** | **0.464** | 0.960 | 0.840 | 0.222 | 0.500 | 0.878 | 0.854 | **0.532** | **0.857** | **0.217** |
| Gemma-2-9b-it | 0.220 | 0.320 | 0.200 | **1.000** | 0.760 | 0.040 | 0.360 | 0.900 | 0.260 | 0.180 | 0.200 | 0.020 |
| LLaMA-3.1-8b-Instruct | 0.000 | 0.235 | 0.157 | 0.408 | 0.500 | 0.020 | 0.580 | 0.820 | 0.420 | 0.000 | 0.360 | 0.200 |
| GPT-4o | **0.451** | 0.569 | 0.294 | **1.000** | **0.940** | **0.824** | 0.549 | **1.000** | **0.863** | 0.255 | 0.588 | 0.176 |
| GPT-3.5-turbo | 0.098 | 0.196 | 0.196 | **1.000** | 0.627 | 0.000 | **0.725** | 0.882 | 0.431 | 0.039 | 0.294 | 0.039 |

High-quality data is essential for effective post-training (Muennighoff et al., 2025; Ye et al., 2025; Li et al., 2025), motivating our focus on careful dataset construction and analysis. Section 4.1 describes our data sources, and Section 4.2 presents baseline experiments with several LLMs on existing benchmarks, analyzing their performance to gain intuitions. Section 4.3 details the creation of the Recon Dataset, while Section 4.4 outlines the distillation of reasoning traces for Recon-CoT.

### 4.1 DATASET SOURCE

We curate four datasets covering distinct facets of economic reasoning: **STEER Benchmark** (Raman et al., 2024) provides ∼600K multiple-choice questions across 48 microeconomic categories, spanning arithmetic, probability, psychological biases, and game theory. Each question includes a prompt, candidate answers, the correct label, and metadata. STEER serves as our primary benchmark for general economic reasoning. **EconLogicQA** (Quan & Liu, 2024) contains 650 human-validated questions inspired by real-world news. Each presents 3–4 interdependent events requiring correct temporal or causal ordering, testing planning and causal consistency. **EconNLI** (Guo & Yang, 2024) offers 11K premise–hypothesis pairs annotated for entailment or neutrality. Derived from Wikipedia, it evaluates a model's ability to infer causal and logical relations in economic narratives. **Pure-Strategy Equilibrium Games** (Fourny & Sulser Larraz, 2020-10-07) consists of 3×3 payoff matrices labeled with Pure Nash and Perfectly Transparent Equilibria. To supplement STEER's noisier game-theoretic items, we convert selected matrices from this ETH Zürich dataset into natural language prompts to assess equilibrium reasoning.

### 4.2 CURATION EXPERIMENT ANALYSIS

Appendix A.5 outlines our dataset curation experiment settings for assessing baseline performance across various LLMs on the collected question types. Our experiment and analysis addresses three key questions: (i) comparative performance of open-weight and closed-source models, (ii) the impact of reasoning distillation, and (iii) identifying specific bottlenecks in economic reasoning skills. Results summarized in Tables 1 and 2 yield the following insights:

**Closed models lead, but reasoning models narrow the gap.** Closed-source GPT-4o consistently achieves top accuracy in most macro-categories, though notably, DeepSeek-R1-Distill-Qwen-7B

slightly surpasses GPT-4o on *Mathematical Foundations* (0.896 vs. 0.887) and *Multi-Agent Environments* (0.630 vs. 0.625). This indicates that specialized open-source reasoning models can effectively rival closed-source proprietary models on fundamental economic tasks.

**Reasoning distillation significantly improves performance.** DeepSeek-R1-Distill-Qwen-7B outperforms all other comparable-sized open models across most macro-categories, particularly excelling in *Multi-Agent Environments*. In contrast to financial domains (Liu et al., 2025b; Qian et al., 2025; Zhu et al., 2025), where R1-style models underperform on commonsense-heavy tasks (e.g., accounting reports), our results suggest that economic reasoning benefits more from structured, multi-step inference. This aligns with Sprague et al. (2025), who find that CoT prompting is most effective on tasks involving symbolic or logical reasoning—helping explain the advantage of System 2 thinking (Kahneman, 2011) in economic, but not financial, domains.

**Complex game-theoretic tasks remain challenging.** Detailed examination in Table 2 reveals significant weaknesses in advanced strategic reasoning, particularly *Trigger strategies* and *Enforceability* in repeated games. Even the leading GPT-4o achieves limited accuracy (0.176 and 0.824 respectively), while most open models fall below baseline on these long-horizon reasoning tasks.

**DeepSeek-R1-Distill-Qwen-7B as the optimal baseline for further training.** Despite these bottlenecks, R1-Distill-Qwen-7B's solid overall performance (macro-average 0.69) and promising baseline competence in strategic reasoning (e.g., 0.217 on *Trigger* and 0.222 on *Enforceability*) make it a strong candidate for subsequent SFT and RL fine-tuning. Its open-source accessibility and manageable scale provide an ideal foundation for enhancing economic reasoning capabilities, particularly in challenging multi-agent contexts.

### 4.3 DATASET CURATION

**Recon Corpus Overview.** We present **Recon Dataset**, emphasizing the 15 most challenging categories identified in our benchmark analysis (Section 4.2). These include advanced game theory, behavioral biases, and logical inference. We curate total 2,100 question-answer pairs: **Training Split (Recon Dataset)**: 1,800 questions, proportionally sampled based on empirical error rates per category (Table 6). **Evaluation Split (Recon-Eval)**: 300 held-out questions (20 per category), mirroring the training distribution.

**Sampling Strategy.** Within each category, we remove ambiguous or low-quality items, then uniformly sample remaining questions to meet predefined quotas (e.g., 250 questions for *Enforceability*, 75 for *Certainty Effect*).

**Prompt Template.** Each question employs a structured prompt that encourages models to reason step-by-step and explicitly box their final answers. A representative example is illustrated in Figure 5.

**Category Breakdown.** Table 6 summarizes the fifteen Recon categories, providing concise descriptions, data provenance, and question counts for the training set. The evaluation set replicates these proportions at one-sixth scale (20 questions per category, totaling 300).

### 4.4 REASONING TRACE DISTILLATION

We distill *chain-of-thought* (CoT) traces from a stronger teacher and filter them for correctness.

**Teacher Prompting.** For each of the 1,800 Recon training items, we issue the same prompt template as in Figure 5 to the teacher model **QwQ-32B** (Team, 2025). The template forces the teacher to put thinking process inside `<think> ... </think>` and to place its final choice in `\boxed{...}` so that both the trace and the answer can be extracted programmatically.

**Filtering.** We parse the teacher's boxed answer and compare it to the gold label. Only items the teacher answers *correctly* are kept. This yields a clean set of **868** (question, gold answer, chain-of-thought) triples covering all 15 Recon categories.

**CoT Corpus.** The resulting 868 demonstrations constitute the **Recon-CoT** dataset. We use this dataset for SFT. As each trace ends with the same extraction-friendly pattern, the fine-tuned model separates reasoning from its verdict, simplifying downstream reward modeling and evaluation.

## 5 MAIN RESULTS

### 5.1 MODEL CONFIGURATION

We conduct our experiments using DeepSeek-R1-Distill-Qwen-7B (Yang et al., 2024; DeepSeek-AI, 2025) as the base model, with all training performed on a single NVIDIA H800 GPU. To enable scalable experimentation, we adopt the Unsloth library (Daniel Han & team, 2023) for memory-efficient fine-tuning and Hugging Face's TRL framework (von Werra et al., 2020) for SFT and RL. For parameter-efficient adaptation, we employ LoRA (rank=8) (Hu et al., 2022) in both SFT and RL. During SFT, we use a batch size of 8 and a learning rate of 2e-4, with a linear learning rate scheduler and 5 warmup steps. For RL, we adopt a batch size of 32, generate 8 samples per optimization step, and apply a cosine learning rate scheduler with an initial rate of 5e-6. We initialize the RL stage from the checkpoint of the SFT-tuned model. Recon-SFT is trained for 2,700 steps. GRPO is then applied for 2,250 steps. See the analysis and plots of training dynamics in Appendix A.6.

### 5.2 EVALUATION DETAILS

We evaluate on three benchmarks: our held-out **Recon-Eval** (300 economic reasoning problems), the **Complete-Information Games** framework (Hua et al., 2024), and **GTBench** (Duan et al., 2024). The Complete-Information Games suite includes 5 simultaneous and 5 sequential games testing agents on (1) communication, (2) cooperation, and (3) strategic alignment. We run 20 trials per game (temperature 0.6, no workflow), where the agent plays *against itself*. Performance is measured by Nash Equilibrium frequency. GTBench focuses on strategic and logical reasoning in competitive settings. Each task is evaluated over 10 trials (temperature 0.6) using PromptAgent *against fixed opponents* (e.g., GPT-4o-mini), with win rate as the metric. For the two gameplay benchmarks, we evaluate four 7B models: Qwen2.5-Instruct, R1-Distill-Qwen, Recon-SFT, and Recon-RL.

### 5.3 ECONOMIC REASONING PERFORMANCE

Table 3 reports accuracy on our 300-item **Recon-Eval** (Section 4.3) set across training stages. Starting from the base model (**48.3%**), SFT boosts performance to **59.7%**, gaining 11.4 points, suggesting that distilled teacher traces effectively transfer structured reasoning patterns. GRPO further improves accuracy to **63.0%**, adding 3.3 points. Overall, the SFT→RL pipeline achieves a **14.7%** absolute gain, validating post-training as a viable strategy for aligning DeepSeek-R1-Distill-Qwen-7B with economic reasoning tasks.

Table 3: Accuracy and percentage score on **Recon-Eval** (evaluated at temperature = 0.0) for the Base Model, Recon-SFT, and Recon-RL.

| Model | Accuracy | Score (%) |
|---|---|---|
| Base Model | 145 / 300 | 48.30 |
| Recon-SFT | 179 / 300 | 59.67 |
| Recon-RL | 186 / 300 | **63.00** |

### 5.4 GENERALIZATION TO STRATEGIC GAMES

To verify that the gains obtained from economic post-training extend beyond single-step reasoning, we evaluate the models in two unseen interactive settings, testing if economic reasoning post-training *generalizes* to strategic interaction.

**Frequent convergence to Nash Equilibria.** Table 4 reveals a clear monotonic gain in self-play equilibrium frequency as economic post-training is added. Relative to the R1-Distill baseline, Recon-SFT increases the proportion of equilibrated outcomes from 0.39 to 0.47 in simultaneous-move games while preserving the strong 0.79 level in sequential games. A subsequent GRPO stage raises these figures to **0.51** and **0.86**, yielding an overall mean of **0.685**, a 9.5 points improvement over R1-Distill and almost double the 0.345 attained by the non-reasoning Qwen-2.5-7B-Instruct.

Table 4: Nash Equilibrium frequency for self-play (higher is better). Settings detailed in Section 5.2.

Table 5: GTBench win rates against GPT-4o-mini (higher is better). Settings detailed in Section 5.2.

| Game | Qwen2.5 | R1-Distill | Recon-SFT | Recon-RL |
|---|---|---|---|---|
| *Simultaneous* | | | | |
| Prisoner's Dilemma | 0.85 | 0.95 | **1.00** | **1.00** |
| Stag Hunt | 0.50 | 0.50 | 0.45 | **0.60** |
| Battle of Sexes | **0.20** | 0.10 | 0.15 | **0.20** |
| Wait-Go Game | 0.15 | 0.25 | **0.70** | 0.65 |
| Duopolistic Competition | **0.15** | **0.15** | 0.05 | 0.10 |
| Avg. (Simul.) | 0.37 | 0.39 | 0.47 | **0.51** |
| *Sequential* | | | | |
| Escalation | 0.15 | 0.95 | 0.85 | **1.00** |
| Monopoly | **0.95** | **0.95** | 0.90 | **0.95** |
| Hot-Cold | 0.05 | 0.65 | 0.90 | **0.95** |
| Draco | 0.40 | 0.75 | 0.75 | **0.90** |
| Trigame | 0.05 | **0.65** | 0.50 | 0.50 |
| Avg. (Seq.) | 0.32 | 0.79 | 0.78 | **0.86** |
| **Overall Avg.** | 0.35 | 0.59 | 0.63 | **0.69** |

| Game | Qwen2.5 | R1-Distill | Recon-SFT | Recon-RL |
|---|---|---|---|---|
| breakthrough | **0.40** | 0.10 | 0.20 | 0.30 |
| connect4 | 0.20 | **0.40** | **0.40** | 0.30 |
| first_sealed_auction | 0.30 | 0.40 | **0.50** | **0.50** |
| kuhn_poker | **0.70** | 0.30 | **0.70** | **0.70** |
| liars_dice | 0.30 | 0.30 | **0.60** | 0.40 |
| negotiation | 0.00 | 0.70 | 0.80 | **0.90** |
| nim | **0.70** | 0.00 | 0.00 | 0.00 |
| pig | **1.00** | 0.90 | 0.50 | 0.80 |
| prisoners_dilemma | 0.00 | **1.00** | **1.00** | **1.00** |
| tictactoe | 0.40 | **0.80** | 0.60 | 0.70 |
| **Overall Avg.** | 0.40 | 0.49 | 0.53 | **0.56** |

Because Nash equilibria embody mutual best responses, more frequent convergence implies the model is better at (i) anticipating the incentives of the other agent and (ii) selecting undominated strategies. We therefore interpret the jump in equilibrium rate as quantitative evidence that post-training injects a *transferable equilibrium prior*: the model has internalized economic rationality principles that apply even to games it never saw during training.

**Economic rationality carries over to competitive play.** The same inductive bias manifests in the strategic game setting. From Table 5, we can observe that Recon-SFT already secures the highest mean win rate among 7B models (0.53). GRPO again provides a consistent lift to **0.56**, winning or drawing in 8 of 10 tasks. The biggest relative gains appear in *negotiation* (+0.20) and *breakthrough* (+0.20), two games that demand extended look-ahead and adaptive bidding abilities never explicitly included in our training corpus. When compared to the non-reasoning model Qwen-2.5-7B-Instruct, the Recon-RL model has a much higher win rate, verifying the idea that reasoning ability helps a model succeed in a strategic game scenario.

Such improvements cannot be explained by pattern memorization or combinatorial search (performance on *nim* is unchanged); instead, they indicate that the economic-reasoning skills learned offline translate into more general strategic behavior against a strong, unseen opponent. The fact that every DeepSeek checkpoint, including Recon-RL, scores low on *nim*, whose solution is a single XOR invariant rather than an incentive-driven best-response problem, underscores this boundary: our post-training injects an equilibrium-seeking bias, not ready-made combinatorial tricks. Thus the miss on *nim* refines our claim that economic post-training chiefly benefits tasks where strategic reasoning, not rote formula recall, is decisive.

## 5.5 Emergent Behaviors from Post-Training in Multi-Agent Games

A qualitative comparison between the Recon-RL and Recon-SFT traces on the *Draco* sequential game (see Figures 6 and 7) reveals several systematic, post-training behaviors:

**Explicit strategic modelling.** Recon-RL spontaneously *constructs the game tree*, labels subgames, and appeals to solution concepts such as "subgame-perfect Nash equilibrium" and "backward induction." Recon-SFT, in contrast, walks through payoff lines informally and never names the underlying equilibrium logic.

**Iterative search and self-correction.** The RL model exposes a lengthy "trial-and-error" chain of thought—simulating each branch, spotting contradictions, and revising intermediate conclusions before converging on the optimal path.

Taken together, these observations suggest that the SFT stage acquires the foundational knowledge for solving the strategic scenarios, while the GRPO stage teaches the model to *simulate the solution procedure* a trained economist would follow, rather than merely memorizing answer patterns. The richer internal search and tighter adherence to formal terminology provide a plausible mechanism for

the quantitative gains reported in Tables 3 and 4 and for the improved win-rates on unseen interactive benchmarks (Section 5.3).

# 6 INSIGHTS AND FUTURE WORK

## 6.1 POST-TRAINING FOR AGENT ALIGNMENT

The jump from *single-shot, textbook* economics to *interactive, adversarial* games in Section 5.4 is striking. We propose two complementary mechanisms that can explain this out-of-domain generalization and discuss their broader implications.

**Structured prompts ⇒ modular latent policies.** The Recon template enforces an explicit *think|act* separation. This mirrors the *inner-rollout / outer-commitment* loop required in game playing: search over hypothetical branches, then output a single move. We conjecture that the template therefore trains a *policy-over-thoughts* module that can be invoked verbatim when the same model is asked to play against another agent, yielding more systematic tree construction and self-correction.

**Outcome-aligned reward ⇒ an "equilibrium prior".** GRPO optimizes a scalar signal that is proportional to *final correctness*. The easiest way for the model to guarantee a non-zero return is therefore to plan *backwards*: select undominated steps that survive any continuation. Over thousands of problems, this trains a bias toward *mutual best responses*. When dropped into a multi-player environment, the same bias manifests as (i) rejecting dominated moves, (ii) gravitating toward equilibrium outcomes.

**Why is this behavior meaningful? Scalable alignment.** Aligning models to "cooperative and rational" behavior usually relies on costly human annotation. Our results indicate that *single-agent, verifiable* datasets already inject a sizable portion of that inductive bias. **Transparency.** The richer, self-correcting chains of thought exposed after GRPO give practitioners a transparent window into the model's decision process, facilitating post-hoc auditing and safety checks.

## 6.2 FUTURE WORK

**Workflow Integration.** We plan to investigate whether integrating multi-agent workflows, such as negotiation and equilibrium resolution frameworks (Hua et al., 2024), can further enhance interactive reasoning and cooperative capabilities.

**Broader Microeconomic Generalization.** We aim to investigate whether post-training on a wider range of microeconomic scenarios—such as bargaining, market clearing, or taxation—can elicit stronger and more stable agentic behaviors.

**Cross-Domain Transfer.** We also aim to investigate whether our post-training approach can generalize beyond economic reasoning to induce other sophisticated aspects of human cognition, such as social cooperation, psychological biases, or ethical decision-making. Demonstrating such broader cognitive generalization would reinforce the potential of domain-aligned post-training as a versatile method for eliciting complex human-like behaviors in language models.

# 7 CONCLUSION

We present **Recon**, a 7B open-source model post-trained for economic reasoning that exhibits strategic generalization. Leveraging a curated dataset of 2,100 problems and a two-stage SFT+GRPO pipeline, Recon achieves a 14.7% improvement on single-agent economic benchmarks and increases Nash equilibrium convergence by 9.5 points in interactive multi-agent games. Our findings suggest that domain-aligned post-training offers a scalable route to economic rationality and induces strategic behavior in previously unseen multi-agent settings.

## LLMs Usage Statement

We clarify that LLMs were used solely as auxiliary tools for paper writing, restricted to two purposes: (i) refining the manuscript's exposition for clarity and conciseness, and (ii) generating preliminary schematic elements for visualizations of methodological pipelines.

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

# A APPENDIX

## A.1 ADDITIONAL RELATED WORK

**Advancements in Large Language Models.** Transformer-based architectures (Vaswani et al., 2017) underpin modern NLP. When scaled to billions of parameters, these models exhibit strong generalization, follow predictable scaling laws (Kaplan et al., 2020; Hoffmann et al., 2022), and demonstrate potential toward AGI (Bubeck et al., 2023; Feng et al., 2024). This scaling has yielded increasingly capable models such as OpenAI o1 (OpenAI, 2024b), Qwen-3 (Yang et al., 2025), LLaMA 4 (Llama Team, 2025), and DeepSeek-R1 (DeepSeek-AI, 2025). To further enhance the efficiency and interpretability of LLMs, researchers have proposed a range of adaptation strategies, including model compression via pruning and quantization (Lin et al., 2024; Kumar et al., 2024; Liang et al., 2025a; Shen et al., 2025c), inference acceleration through layer skipping (Shen et al., 2025a;b), data distillation (Zhao et al., 2020; Wang et al., 2024; 2025d; Guo et al., 2024; Wang et al., 2018; 2025c), data selection (Xia et al., 2024a;b; Wang et al., 2025a; Xu et al., 2025b), token pruning and merging (Wang et al., 2025b; Bolya et al., 2022; Wen et al., 2025; Liu et al., 2025a; 2024), theoretical analyses of representation power (Chen et al., 2025b;c; Liang et al., 2025b), and advances in interpretability and mechanistic understanding (Bereska & Gavves, 2024; Wang et al., 2025b; Covert et al., 2022). These developments underscore the increasing importance of understanding, optimizing, and governing the behavior and performance of large language models.

## A.2 SUPERVISED FINE-TUNING

Supervised Fine-Tuning (SFT) adapts pretrained models to specific tasks by imitating input–output pairs, effectively aligning models with structured reasoning when high-quality demonstrations are available (Muennighoff et al., 2025; Ye et al., 2025). We distill outputs from strong reasoning models (e.g., DeepSeek-R1 (DeepSeek-AI, 2025), QwQ-32B (Team, 2025)) containing both reasoning traces and final answers, training the model to generate solutions with coherent thought processes.

Let $\mathcal{D}_{\text{SFT}} = \{(x_i, y_i)\}_{i=1}^N$ denote the fine-tuning dataset of size $N$, where $x_i$ is the input prompt and $y_i$ is the target output. In our formulation, each output is a tuple $y_i = (c_i, a_i)$, where $c_i$ represents the step-by-step reasoning and $a_i$ the final answer. The training objective minimizes the negative log-likelihood of the output tokens:

$$\mathcal{L}_{\text{SFT}}(\theta) = -\mathbb{E}_{(x,y)\sim\mathcal{D}_{\text{SFT}}}\left[\log p_\theta(y \mid x)\right],$$

where $\theta$ are the model parameters and $p_\theta(y \mid x)$ is the conditional probability of generating $y$ given $x$. The loss is computed only over the output tokens $y$, excluding the prompt $x$.

To reinforce structured reasoning, we standardize the output format by enclosing the reasoning process $c_i$ within special `<think>` and `</think>` tokens. This explicit markup helps the model distinguish intermediate steps from final outputs and provides structure that is beneficial for downstream reward modeling. Overall, SFT provides a strong initialization that enhances reasoning generalization and stabilizes subsequent RL stage.

## A.3 GRPO

We adopt Group Relative Policy Optimization (GRPO) (Shao et al., 2024) as our reinforcement learning post-training algorithm. GRPO improves efficiency by eliminating the need for a value function and instead estimates advantages from a group of sampled outputs. For each input query $q$, drawn from the data distribution $\mathcal{D}_q$, a group of $G$ responses $\{o_1, o_2, \ldots, o_G\}$ is sampled from the old policy $\pi_{\theta_{\text{old}}}$. The current policy $\pi_\theta$ is then optimized by maximizing the following objective:

$$\mathcal{J}_{\text{GRPO}}(\theta) = \mathbb{E}_{q\sim\mathcal{D}_q, \{o_i\}_{i=1}^G \sim \pi_{\theta_{\text{old}}}(\cdot|q)}\left[\frac{1}{G}\sum_{i=1}^G \min\left\{w_i A_i, \text{clip}(w_i, 1-\epsilon, 1+\epsilon)A_i\right\} - \beta\,\text{KL}(\pi_\theta\|\pi_{\text{ref}})\right],$$

where $w_i := \frac{\pi_\theta(o_i|q)}{\pi_{\theta_{\text{old}}}(o_i|q)}$.

Here, $\pi_\theta(o_i|q)$ denotes the probability of generating response $o_i$ given query $q$ under the current policy, and $\pi_{\theta_{\text{old}}}(o_i|q)$ is the probability under the old policy used for sampling. The advantage $A_i$

reflects the relative quality of each response and is computed as the normalized reward within the group:

$$A_i = \frac{r_i - \text{mean}(\{r_1, r_2, \ldots, r_G\})}{\text{std}(\{r_1, r_2, \ldots, r_G\})},$$

where $r_i$ is the scalar reward assigned to output $o_i$. The KL penalty encourages stability by penalizing deviations from a reference policy $\pi_{\text{ref}}$, and is defined as:

$$\text{KL}(\pi_\theta \| \pi_{\text{ref}}) = \frac{\pi_{\text{ref}}(o_i|q)}{\pi_\theta(o_i|q)} - \log \frac{\pi_{\text{ref}}(o_i|q)}{\pi_\theta(o_i|q)} - 1.$$

The hyperparameter $\epsilon$ controls the clipping threshold for policy ratio updates, while $\beta$ scales the KL regularization strength.

### A.4 DETAILED REWARD DESIGN

To align model outputs with structured behavior, we design a hierarchical, rule-based reward function for GRPO training, inspired by DeepSeek-R1 (DeepSeek-AI, 2025). The reward evaluates each response across three stages: structural formatting, parseability, and correctness.

We follow DeepSeek's usage guidance[2] by prepending a `<think>` token to each response, prompting the model to first generate a reasoning trace, followed by a boxed final answer. Omitting this token degrades both coherence and accuracy.

Final answers are extracted via string matching, with a strong preference for `\boxed{}` formatting. To address formatting inconsistencies in the Qwen family, we penalize deviations from the expected structure, encouraging alignment between reasoning and final predictions. Our reward design is illustrated for multiple-choice example questions (Figure 5), where answers follow the format `\boxed{Option X: full choice text}`.

Formally, our hierarchical reward function comprises three stages:

- **Stage-A (Format Check):** Each response must contain exactly one `</think>` tag, and any `\boxed{}` answer must appear afterward. Violations of these constraints incur a format penalty.
- **Stage-B (Answer Extraction):** We attempt to extract the first boxed answer appearing after `</think>`. If unavailable, we fallback to the first occurrence of an alternative format such as `Option X`. Inability to extract any answer incurs a parse penalty.
- **Stage-C (Correctness Grading):** If the extracted answer exactly matches the reference answer, we assign a high positive reward. A partial reward is given if only the option number matches (e.g., both indicate "Option 2"). Incorrect answers receive a negative penalty.

The explicit reward values assigned are summarized as follows. Let $r(o)$ denote the reward for a model output $o$:

$$r(o) = \begin{cases} +5 & \text{exact match,} \\ +2 & \text{partial match,} \\ -3 & \text{incorrect answer,} \\ -4 & \text{format violation,} \\ -5 & \text{parse failure.} \end{cases}$$

This hierarchical, rule-based scoring framework ensures determinism, interpretability, and efficient reward signal propagation, effectively supporting the acquisition of correct economic reasoning behaviors during RL post-training.

### A.5 CURATION EXPERIMENT SETTING

**Models.** We evaluate six models: closed-source **GPT-4o** (OpenAI, 2024a), **GPT-3.5-Turbo** (OpenAI, 2022); and open-weight **DeepSeek-R1-Distill-Qwen-7B** (DeepSeek-AI, 2025), **Qwen-2.5-7B-Instruct** (Yang et al., 2024), **Llama-3.1-8B-Instruct** (Grattafiori et al., 2024), and **Gemma-2-9B-It** (Team et al., 2024).

---

[2]https://huggingface.co/deepseek-ai/DeepSeek-R1-Distill-Qwen-7B#usage-recommendations

**Question Pool.** We sample 50 questions per category from STEER (48 categories) (Raman et al., 2024), and 50 each from EconLogicQA (Quan & Liu, 2024) and EconNLI (Guo & Yang, 2024), yielding a 2,500-question pool spanning 50 categories. These are grouped into five macro-categories:

- **Mathematical Foundations**: Tests whether a model can handle the "nuts-and-bolts" of economic analysis: basic arithmetic, optimization under simple constraints, probability calculations, and short chains of deductive logic. Typical items range from computing a quick sum in *add_sub* to working out an expected value in *compute_expectations*.
- **Single-Agent Environments**: Casts the model as a lone decision-maker who weighs costs, benefits, and risk. Besides testing the classical Von Neumann–Morgenstern axioms, the block challenges the model to sidestep behavioral pitfalls such as the *sunk_cost* fallacy or the *endowment_effect*.
- **Multi-Agent Environments**: Shifts to strategic settings where pay-offs depend on how others act. Questions may ask for the optimal first move in a sequential game (*backward_induction*) or for designing a punishment scheme that sustains cooperation in an infinitely repeated game (*trigger* strategies).
- **Representing Other Agents**: Treats the model as a social planner or mechanism designer who must aggregate many individual preferences into a single decision. Examples include checking whether a social ranking is Pareto efficient (*pareto_sc*) or selecting the winner under a simple *plurality_voting* rule.
- **Logical Reasoning**: Adds domain-specific deductive tests from EconLogicQA and EconNLI, asking the model to order socioeconomic events coherently or decide whether one economic event logically entails another.

The categories—Mathematical Foundations, Single-Agent Environments, Multi-Agent Environments, and Representing Other Agents—are sourced from STEER. Logical Reasoning is derived from EconLogicQA and EconNLI.

**Drilling down on Multi-Agent Environments.** Table 2 unpacks the single "Multi-Agent" bar into its five game-theoretic sub-modules so we can clearly see *where* the models trip up.

- **Normal-form games**: simultaneous-move interactions presented as payoff matrices. Items range from picking a dominant/dominated strategy (*Dom. Strat.* and *Dom'd Strat.*) to computing a pure-nash equilibrium.
- **Extensive-form games**: sequential play laid out as game trees. Typical questions ask for the optimal first action via backward induction *(Back. Ind.)* or for identifying a subgame-perfect Nash *(Subg. Nash)*.
- **Infinitely repeated games**: long-horizon interactions that hinge on credible punishment. Here, the model must judge feasibility *(Feas.)*, enforceability *(Enf.)*, or design a trigger *(Trig.)* strategy.
- **Bayesian games**: strategic choice when pay-offs depend on hidden types; the flagship task is computing a *Bayes-Nash* equilibrium.

These finer-grained results intend to reveal a universal weakness in advanced game theory concepts, especially regarding extensive-form games and infinitely repeated games.

**Inference Protocol.** Open models are run via `vLLM` (Kwon et al., 2023) on identical NVIDIA T4G Tensor Core GPU instances; closed models via OpenAI API. Each prompt requests an answer enclosed in `\boxed{...}` plus free-form reasoning. An example question prompt when querying the tested models is shown in Figure 4.

**Evaluation Metric.** A response is marked correct if the boxed answer matches the gold label. Accuracy is computed via exact match. Table 1 summarizes results by macro-category. Table 2 delves into the results for Multi-Agent Environments category.

A.6 TRAINING DYNAMICS

Figure 3 illustrates that the SFT loss decreases steadily and converges smoothly, whereas GRPO reward trends upward and stabilizes around a positive mean, suggesting effective alignment and

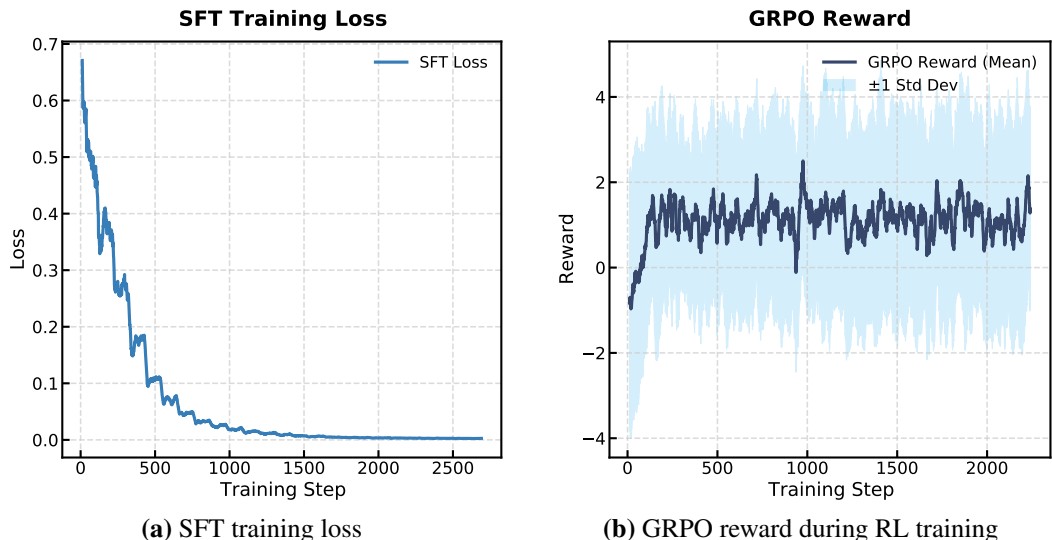

**(a)** SFT training loss       **(b)** GRPO reward during RL training

Figure 3: Training dynamics for SFT (a) and RL (b).

successful reward-guided optimization despite early variance. Furthermore, we observe that SFT provides essential economic knowledge and reasoning priors that enable stable RL optimization. In contrast, our attempts to run RL directly from the base model, despite careful tuning, did not yield full convergence. This underscores the significance of SFT as a vital warm-start, particularly in domains beyond mathematics and coding.

---

**Example Dataset Curation Experiment Question Prompt**

Question:
In a high-stakes acquisition, two competing investors, Alex and Taylor, are negotiating with a major corporation for exclusive rights. The negotiations are structured over three rounds, where each investor alternates in making decisions. Alex makes the first move. If Alex finalizes the negotiation in the first round, they will secure a profit of 31709.805571865647 and Taylor will receive 70026.13028485939. Should Alex choose to continue the negotiations, Taylor can either decide to finalize in the second round or push the decision back to Alex. If Taylor chooses to finalize in the second round, Alex will receive 36394.29786932465 and Taylor will secure 47402.72116860709. If Taylor decides against finalizing and returns the decision to Alex, Alex can choose to end the negotiations with the agreed terms, or let the corporation impose their final terms. If Alex decides to finalize, their profit would be 99028.19689989614 while Taylor's would be 83676.14380026297. If the corporation ends up setting the final terms, Alex will receive 99028.19689989614 and Taylor will get 83676.14380026297. Given this scenario, what should be Alex's strategy in the very first round?

Options:
Option 1: Alex should finalize the negotiation
Option 2: Alex should continue and pass the negotiation to Taylor

Choose the correct option and explain your reasoning. Please reason step by step, and put your final answer within \boxed{}.

---

Figure 4: Example question prompt for *backward_induction* used in the Dataset Curation Experiment.

Table 6: Categories, Distributions, and Descriptions of Recon Training Dataset

| Class | Description | Source | Count | Proportion |
|---|---|---|---|---|
| Enforceability | Incentive compatibility in long-term relationships | Repeated Game | 250 | 13.9% |
| Backward Induction | Optimal choice in sequential decisions | Sequential Game | 250 | 13.9% |
| Trigger | Punishment-based strategies to enforce cooperation | Repeated Game | 250 | 13.9% |
| Feasibility | Sustainability of payoff allocations | Repeated Game | 150 | 8.3% |
| Auction Risk | Risk preferences in bidding contexts | One-shot Game | 150 | 8.3% |
| Endowment Effect | Overvaluation of owned assets | Behavioral | 75 | 4.2% |
| Certainty Effect | Preference for guaranteed outcomes over probabilistic ones | Behavioral | 75 | 4.2% |
| Time Inconsistency | Dynamic inconsistency in intertemporal choices | Behavioral | 25 | 1.4% |
| Budget Balance | Financial balance in risk-sharing settings | Risk Management | 50 | 2.8% |
| Condorcet Criterion | Majority rule consistency in voting | Majority Vote | 25 | 1.4% |
| Bayes Nash | Strategic reasoning under probabilistic uncertainty | Probability | 50 | 2.8% |
| EconLogicQA | Stepwise logical reasoning in economics | Logic | 150 | 8.3% |
| EconNLI | Economic causal inference in natural language | Logic | 100 | 5.6% |
| Pure Nash | Existence and identification of pure strategy equilibria | Game Theory | 100 | 5.6% |
| PTE | Perfect Transferable Equilibrium decision logic | Game Theory | 100 | 5.6% |
| **Total** | | | **1800** | **100%** |

---

**Example Recon Training Dataset Question Prompt**

You are given a reasoning problem in Economics. Please reason step by step to solve it.

Then, give your final answer using the format \boxed{Option X: full choice text}.

You must include both the option number and the full text of your chosen answer.

Question:
Consider a duopoly where two firms, A and B, sell a homogeneous product. Each firm can choose between undercutting the competitor's price (compete) or colluding to maintain a high price (collude). In each period, if both firms compete, each earns $43334.64; if both collude, each earns $98183.38; if firm A competes while firm B colludes, A earns $77043.33 and B earns $56330.53. Conversely, if firm B competes while A colludes, B earns $77043.33 and A earns $56330.53. Assuming firms discount future payoffs with a discount factor of 0.44, which of the following strategies can be sustained as an enforceable outcome in a Nash equilibrium of an infinitely repeated game?

Options:
Option 1: Firms A and B compete on price in each period indefinitely.
Option 2: Firms A and B collude to maintain high prices in each period indefinitely.
Option 3: Firms A and B take turns competing and colluding in alternating periods.
Option 4: Firm A competes and Firm B colludes indefinitely.

Figure 5: Example question prompt used in the Recon training dataset.

> **Example Draco Game Output of Recon-RL**
>
> Okay, so I'm trying to figure out what choice Alice should make to maximize her reward in this game. Let me start by understanding the rules thoroughly.
> From the game rules, it seems like this is a **sequential game** where Alice and Bob take turns making choices. The rewards depend on the sequence of choices they make. Let me break down each scenario...
>
> Hmm, so it's a bit complex because the rewards depend on the sequence of choices. Alice has to consider Bob's possible responses and her own subsequent choices.
>
> Let me try to represent this as a **game tree** to visualize the possible outcomes...
>
> Now, I need to analyze the optimal strategies for both players. Since it's a **sequential game**, we can use **backward induction** to find the **subgame perfect Nash equilibrium**...
>
> Let's start from the end of the game and work our way back...
>
> Wait, no. Let me correct that. If Bob chooses $choice_2$, he gets 2, but Alice can get 3 or 2. Wait, actually, the payoffs are from Alice and Bob's perspective.
>
> Wait, in Subgame 1: Alice chooses $choice_1$, Bob chooses $choice_2$, then Alice chooses $choice_1$: Alice gets 2, Bob gets 2...
>
> Wait, that's conflicting. Let me clarify...
>
> So, comparing Bob's payoffs: - If Bob chooses $choice_1$: Bob gets 5.
> - If Bob chooses $choice_2$: Bob gets 3.
>
> Therefore, Bob's best response is to choose $choice_1$ because 5 ¿ 3.
>
> So, if Alice chooses $choice_2$, Bob will choose $choice_1$, resulting in Alice getting 4 and Bob getting 5.
>
> Now, let's summarize:
> - If Alice chooses $choice_1$: Both get 5.
> - If Alice chooses $choice_2$: Alice gets 4, Bob gets 5.
>
> Comparing these, Alice would prefer to choose $choice_1$ to get 5 instead of 4.
>
> Therefore, Alice's optimal choice is $choice_1$.

Figure 6: Example Draco game (Hua et al., 2024) output of Recon-RL. Some important game theory terminology employed by the model is bolded for reference.

1296
1297
1298
1299
1300
1301
1302
1303
1304
1305
1306
1307
1308
1309
1310
1311
1312
1313
1314
1315
1316
1317
1318
1319
1320
1321
1322
1323
1324
1325
1326
1327
1328
1329
1330
1331
1332
1333
1334
1335
1336
1337
1338
1339
1340
1341
1342
1343
1344
1345
1346
1347
1348
1349

---

**Example Draco Game Output of Recon-SFT**

Alright, so I'm trying to figure out the best move for Alice in this **sequential game**. Let me break it down step by step.

First, I'll list out all the possible scenarios and their corresponding rewards. It might help to visualize it as a tree or a table...

Hmm, so it's a bit complex because there are multiple rounds here. Let me see if I can figure out the optimal strategy for both players...

Starting with the first move, Alice has to choose either $choice_1$ or $choice_2$ without knowing Bob's choice. So, let's consider each possibility...

Now, considering that both players are trying to maximize their rewards, they'll probably anticipate each other's moves...

Wait, no. If Alice chooses $choice_2$, Bob can choose $choice_1$, which gives him 5, which is better than 3. So Bob would definitely choose $choice_1$ if Alice chooses $choice_2$ first...

But wait, if Alice chooses $choice_1$, Bob can choose $choice_2$, leading to a two-round game. In that case, Alice can get either 2 or 3. But 3 is better than 2, so she would choose $choice_2$, resulting in Alice:3, Bob:4...

But wait, Bob can choose $choice_2$ if Alice chooses $choice_1$, leading to a lower reward for Alice. So, maybe Alice should consider that Bob will try to minimize her reward...

Wait, but if Alice chooses $choice_2$, Bob gets 5. If Alice chooses $choice_1$, Bob can get 5 as well. So, Bob's reward is the same in both cases, but Alice's reward is higher if she chooses $choice_2$...

Wait, that changes things. So, if Alice chooses $choice_1$, Bob's best response is to choose $choice_1$, resulting in both getting 5...

If Alice chooses $choice_2$, Bob chooses $choice_1$, giving Alice 4 and Bob 5.

So, Alice can get either 5 or 4, depending on her choice.

Since 5 is better than 4, Alice should choose $choice_1$.

Therefore, the best move for Alice is to choose $choice_1$, leading to both players getting 5.

Figure 7: Example Draco game (Hua et al., 2024) output of Recon-SFT. Some important game theory terminology employed by the model is bolded for reference.

