# OpenReview forum: "Strategic Generalization Without Interaction: Can Post-Training Alone Induce Multi-Agent Behavior?"
_ICLR.cc/2026/Conference — ICLR 2026 Conference Withdrawn Submission_

### Official Review · Reviewer_FWn1 · 2025-10-26

**Soundness:** 2
**Presentation:** 2
**Contribution:** 3
**Rating:** 4
**Confidence:** 3

**Summary:**

The authors introduce Recon, a 7B model (DeepSeek‑R1‑Distill‑Qwen‑7B base) post‑trained on a curated 2,100‑item economic/game‑theory dataset. They do SFT on 868 teacher CoT traces and then GRPO on 1,800 items with a rule‑based reward (format + parse + correctness). They report A) +14.7 points on their 300‑item eval set (“Recon‑Eval”), B) higher Nash equilibrium frequency in self‑play across 10 games, and C) higher win rates on GTBench (10 tasks) against a fixed opponent. They interpret these as evidence that domain‑aligned post‑training yields “equilibrium‑seeking” behavior in multi‑agent settings despite the absence of gameplay trajectories during post‑training.

**Strengths:**

- This paper asks a great question (though it's not phrased properly imo, see below): if you only do single-turn, non-interactive post-training in a well-structured domain (here, economics), do you get any lift in interactive multi-agent games? This is a very good research direction and well-chosen impactful question, worth exploring before jumping to more complicated MAS work as others are doing.
- The authors give a concrete, reproducible recipe (SFT -> GRPO on a 7B). I also like several of the results:
    - On the held-out Recon-Eval set you see the big expected jump from SFT (48.3 -> 59.7), with a smaller nudge from RL (reported 63.0, although 186/300 is 62.0).
    - In interactive settings, self-play NE frequency rises from 0.59 (R1-Distill) to 0.63/0.69 (Recon-SFT/RL), and win-rate vs GPT-4o-mini climbs from 0.49 to 0.53/0.56. Those are directionally consistent across two suites, and several per-game jumps (e.g., Wait-Go, Hot-Cold, negotiation) look meaningfully better than noise (though as I'll mention below I'm having to eyeball that a bit)
    - I like the trace-level evidence: the model more often naming the right solution concepts (“subgame-perfect Nash equilibrium,” “backward induction”), and the chains reading more like deliberate search—simulate branches, spot contradictions, revise, and only then commit. This is an interesting approach to improving multi-agent performance.

**Weaknesses:**

- The headline question is ill-posed as written (“Can post-training alone induce multi-agent behavior?”). Of course it can, e.g if your post-training corpus contains multi-agent traces, you’ll learn multi-agent behavior. What the paper really studies is narrower and more interesting: whether single-agent, non-interactive post-training carries over to interactive play.
- Relatedly, the paper over-reaches on claims. The evidence supports near-domain transfer (you teach repeated-game enforceability, trigger strategies, backward induction in single-turn form; performance improves when those same concepts appear in dialogue). Calling this “emergent multi-agent behavior” is premature: the training directly targets the solution concepts used at test time, just in a different format. And the jump from “equilibrium-seeking” to “alignment” is too quick - e.g., in Prisoner’s Dilemma, perfect NE convergence is mutual defection, not cooperative alignment. What do you mean by "agent alignment"? Is this referring to LLM alignment in the broader sense?
- Given your non-interactive -> interactive results claim is strong, results have quite a high bar to clear that I feel isn't quite met. Interactive results use 20 trials/game (self-play) and 10 trials/task (GTBench). The overall averages move in the right direction, but some per-task swings (e.g., liar’s dice up then down) signal variance. Needs confidence intervals, multiple seeds, and a cross-play matrix (Recon-RL vs R1-Distill, vs earlier Recon checkpoints) so we can rule out self-play artifacts and seed luck. Also fix the small arithmetic slip (186/300 = 62.0%, not 63.0%).
- Finally, if “emergent strategic behavior” is going to be a centerpiece, I’d like more detail / convince me more on some of these trace-level details. I'm currently a bit skeptical. Some rough ideas to give you a better sense of what I mean here:
    - Quantify the behavior change in traces. How often does the model explicitly invoke formal concepts (SPNE, undominated strategies, backward induction) pre- vs post-RL? What’s the average branch depth explored, number of reversals/self-corrections, and time-to-commit?
    - Do these correlate with NE convergence or win-rate? Is this just from the SFT? is this actually emergent behavior?
    - Ablate the template.
    - Broaden the interactive tests. Add public-goods games, auctions with private signals, 3+ player settings, or other games not directly represented in the single-turn curriculum.

In short: I like the question, I like the general direction of the results, and I like that your traces look more “economist-like” and that as a path to improving multi-agent performance. But the framing needs tightening, the claims need toning down, and I'd like to see more results with CIs, plus trace-level quantification of the emergent behaviors you describe.

**Questions:**

Please address questions in weaknesses section

---

### Official Review · Reviewer_vMuW · 2025-10-26

**Soundness:** 2
**Presentation:** 3
**Contribution:** 2
**Rating:** 4
**Confidence:** 3

**Summary:**

This paper aims to investigate whether post-training an LLM on economic datasets can enhance its ability to engage in multi-agent interactions when applied to various game-theoretic scenarios, particularly in unseen games. To this end, the authors first curated a high-quality dataset from different benchmarks in the economic area, which is used for SFT on the 7B LLM. Then, the authors used GRPO to further enhance the reasoning capability of the SFT-tuned model. Finally, the authors conducted extensive experiments to demonstrate that the SFT-tuned model can effectively learn the structured reasoning patterns from the teacher traces, and the GRPO (the RLVR method) can further boost the model's performance on reasoning in different unseen strategic games.

**Strengths:**

- The curated dataset is valuable and could have a good impact on the related community.

- The experiments are well-structured and well-executed, clearly supporting the conclusions in the paper.

**Weaknesses:**

- From my understanding, the key contributions of the paper are two-fold: the curated dataset and the findings from experiments.  For the methodology part, no new techniques are proposed.  Thus, I feel this paper is more suitable for a benchmark paper, not a paper with technical innovations.

- Given that the paper is focused on experiments, a more elegant organization of the paper is necessary to show the findings from the experiments. In fact, though the average performance shows an increasing trend in different games, the model can perform worse after using SFT and GRPO in some games. A deeper investigation into such cases is important to show when post-training can succeed in different games.

**Questions:**

See **Strengths** and **Weaknesses**.

---

### Official Review · Reviewer_Ak24 · 2025-10-31

**Soundness:** 2
**Presentation:** 2
**Contribution:** 2
**Rating:** 2
**Confidence:** 3

**Summary:**

This paper investigates whether post-training techniques (Supervised Fine-Tuning and Reinforcement Learning) can enable Large Language Models to generalize from single-agent economic reasoning tasks to multi-agent strategic scenarios without any interactive gameplay data. The authors introduce Recon, a 7B parameter model post-trained on 2,100 curated economic reasoning problems.

**Strengths:**

1.	A curated dataset spanning 15 economic reasoning categories, focusing on challenging areas
2.	Demonstration that SFT followed by GRPO training improves economic reasoning benchmark accuracy by 14.7%
3.	Evidence of emergent strategic behavior: the post-trained model achieves higher Nash equilibrium convergence (+9.5%) in self-play and improved win rates (+7%) against opponents in unseen multi-agent games

**Weaknesses:**

1.	The curation of the datasets seems arbitrary without sufficient justifications. The curation process selects the data from existing benchmarks, STEER, EconLogicQA, EconNIL, and Pure-strategy equilibrium games. Why these benchmarks? Why no others? Does this is sufficient for the coverage? Any dataset bias? Or does these datasets include the nash equilibrium relevant strategies, which will benefit the experiments? If so, the strategic behaviors are not emerging, they are supervised.
2.	The limited technical contributions of the training recipe. The SFT+GRPO is a standard training recipe for LLMs. Is there any technical contributions?
3.	The limited evaluation of the methods. Small sample sizes (10-20 trials per game) without confidence intervals or significance tests. Win rates against a single opponent (GPT-4o-mini) and no comparison with prompting strategies. No ablation studying SFT-only vs. RL-only vs. SFT+RL.

**Questions:**

1.	I want the authors to have some clarifications. “Can post-training techniques generalize effectively to multi-agent scenarios? ” What exactly this mean? This means achieving good performance in games (focusing one player in multi-agent cases) or predicting nash equilibrium (focusing multiple players in multi-agent cases)? If focusing on one player, what is the difference compared with sing-agent cases, you can directly train the LLMs to play Nash equilibrium strategy with SFT. If focusing on multiple players, there are cooperative cases, competitive cases and mixed cooperative and competitive cases, which are totally ignored by this work. So please elaborate more about this.
2.	As the point 1 in the weakness, why choosing such curation process of the datasets? it seems arbitrary, and not all games are related to econ, so this is a bit weird. From my personal perspective, this is just some reasoning data and not much different compared with math or code. So what is the uniqueness of such datasets and such curation process?
3.	As the point 2, what is your technical contributions? Is it just merely training the LLMs over some new data? If so, I do not think this is a good technical paper.
4.	Your evaluation is far from satisfactory. Game is highly stochastic, you should enumerate all possible game paths for some games, e.g., connect 4. Win rates should consider more opponents, so why not GPT-4o or DeepSeek-R1, and even prompting strategies as baselines. Why SFT+RL? How about RL only?

---

### Official Review · Reviewer_zYzW · 2025-11-01

**Soundness:** 2
**Presentation:** 3
**Contribution:** 2
**Rating:** 4
**Confidence:** 4

**Summary:**

The paper studies whether post-training without any interactive multi-agent data can induce multi-agent behavior in LLMs. The authors curate a dataset of 2100 economic problems and introduce Recon to post-train a 7B model through SFT and GRPO. Experiment results show that Recon improves performance on a held-out eval set, and generalizes to interactive multi-agent games, including some complete-information games, and GTBench.

**Strengths:**

1. Clear presentation and analysis: the core thesis is stated clearly, and the paper combines quantitative results with qualitative reasoning traces to support claims.
2. Good empirical improvements. Recon shows consistent gains on the curated econ evaluation set and reports improvements in self-play and on GTBench.

**Weaknesses:**

1. Lack of comparative baselines against other reasoning datasets: many existing works, such as DeepSeek-R1 [1], have already shown that post-training on reasoning datasets (e.g., math, coding) can generalize to unseen scenarios. To prove that the capability improvement truly stems from the economic dataset, the paper should include control post-training runs on established math/coding datasets and compare transfer to the same multi-agent tasks. These baselines would clarify whether economics-specific supervision is necessary or merely sufficient.
2. Limited scale of the post-training dataset: the dataset has only 2100 problems in total with 868 for SFT, which is a very small dataset for post-training a 7B model. For comparison, deepscaler [2] uses 40k samples for training a 1.5B model, and DeepSeekMath [3] uses 776k samples for SFT and 144k samples for RL with a 7B model. A larger-scale dataset would be of greater value for future research in this domain and would lead to more robust model improvements.
3. Insufficient rigor in multi-agent evaluation:
    1. Low trial count and high variance: the number of evaluation trials is very low (20 trials for Complete-Information Games, 10 for GTBench). Given the high variance inherent in multi-agent interactions, these small sample sizes make it difficult to draw reliable conclusions or rule out stochasticity. Reporting variance or other statistical measures over more repeated experiments would be necessary to confirm the robustness of the results.
    2. Limited opponent diversity: multi-agent challenges go beyond finding Nash equilibria and include non-transitive dynamics (i.e., 'rock-paper-scissors' cycles) [4]. Evaluating only against a single opponent (GPT-4o-mini in GTBench) or against oneself (self-play in Complete-Information Games) does not fully demonstrate the model's capabilities. A 'cross-play' evaluation against other models in the suite, and testing against more varied opponents in GTBench, would provide stronger proof of general multi-agent improvement.
    3. Limited environments: the paper's evaluations focus almost exclusively on competitive or mixed-motive game-theoretic settings. However, a major part of multi-agent interaction involves cooperative scenarios, such as problem-solving, software development, or mathematical reasoning (e.g., in systems like ChatDev [5] or AutoGen [6]). The evaluation is missing this entire class of cooperative multi-agent tasks.

[1] Shao, Zhihong, et al. “DeepSeek-R1: Incentivizing Reasoning Capability in Large Language Models via Reinforcement Learning.” arXiv preprint arXiv:2501.12948, 2025.

[2] Li, Beibin, et al. “DeepScaler: Holistic Autoscaling for Microservices Based on Spatiotemporal GNN with Adaptive Graph Learning.” arXiv preprint arXiv:2309.00859, 2023.

[3] Shao, Zhihong, et al. “DeepSeekMath: Pushing the Limits of Mathematical Reasoning in Open Language Models.” arXiv preprint arXiv:2402.03300, 2024.

[4] Czarnecki, Wojciech Marian, et al. “Real World Games Look Like Spinning Tops.” Advances in Neural Information Processing Systems (NeurIPS) 33, 2020, arXiv:2004.09468.

[5] Qian, Chen, et al. “ChatDev: Communicative Agents for Software Development.” arXiv preprint arXiv:2307.07924, 2023.

[6] Wu, Qingyun, et al. “AutoGen: Enabling Next-Gen LLM Applications via Multi-Agent Conversation Framework.” arXiv preprint arXiv:2308.08155, 2023.

**Questions:**

1. Can you report results for post-training on math and coding reasoning datasets (keeping model/backbone and steps comparable) to isolate the specific contribution of econ supervision to multi-agent transfer?
2. More rigorous multi-agent evaluation. Could you provide additional episodes, multiple seeds, opponent diversity (cross-play/populations), and cooperative tasks (e.g., AutoGen/ChatDev/CAMEL-style setups), as noted in Weakness 2?
3. Given that GPT-4o appears strong in your baseline analysis, did you consider distilling from GPT-4o instead of (or in addition to) QwQ-32B? Do you expect a stronger teacher to yield a larger/better Recon-CoT corpus and correspondingly higher SFT performance?

---

### Note · Authors · 2025-12-28

I have read and agree with the venue's withdrawal policy on behalf of myself and my co-authors.